# Structure of the human P2X3 receptor reveals the basis for subtype-selective inhibition by sivopixant

Zhixuan Zhao[1◐], Dong-Ping Wang[2◐], Xin Zhang[2], Yuan Gao[2], Hexin Xu[1], Xinyu Teng[1], Cheng Shen[1], Jirui Chen[1], Jinru Zhang[3], Chang-Run Guo[2]*, Motoyuki Hattori[1]*

1 State Key Laboratory of Genetics and Development of Complex Phenotypes, Collaborative Innovation Center of Genetics and Development, Department of Physiology and Neurobiology, School of Life Sciences, Fudan University, Shanghai, China, 2 State Key Laboratory of Natural Medicines, School of Basic Medicine and Clinical Pharmacy, China Pharmaceutical University, Nanjing, China, 3 State Key Laboratory of Genetics and Development of Complex Phenotypes, Department of Biochemistry and Biophysics, School of Life Sciences, Fudan University, Shanghai, China

◐ These authors contributed equally to this work.
* gcr@cpu.edu.cn (C-RG); hattorim@fudan.edu.cn (MH)

## Abstract

P2X receptors are ATP-gated cation channels, and the P2X3 subtype plays crucial roles in peripheral sensory neurons, including in chronic pain and chronic cough. Accordingly, P2X3 receptors have attracted substantial interest as a therapeutic target. Gefapixant, a negative allosteric modulator (NAM) of P2X3 receptors, has been approved in some countries for the treatment of chronic cough; however, its limited selectivity for P2X3 homomers over P2X2/P2X3 heteromers is associated with taste disturbance as a prominent adverse effect. These limitations have motivated the development of next-generation NAMs with improved subtype selectivity, but their subtype-specific allosteric inhibition mechanisms are unclear. Here, we report the cryo-EM structure of the human P2X3 receptor in complex with ATP and the P2X3-selective next-generation NAM sivopixant, an investigational drug. Sivopixant binds to an allosteric site at the portal of the central pocket in the extracellular domain, and structure-based mutational analysis by electrophysiology identifies key residues required for sivopixant-dependent inhibition of human P2X3 receptors. Structural comparisons across P2X subtypes, together with patch-clamp analyses of gain-of-function mutants that confer sensitivity to two investigational drugs, sivopixant and camlipixant, provided a broadly applicable structural framework for subtype selectivity. Furthermore, structural comparisons with apo and ATP-bound open states of P2X3 receptors, together with molecular dynamics simulations, revealed that sivopixant expands the upper-body domain to suppress the lower-body movements required for channel activation, thereby preventing channel opening even in the presence of ATP.

**Data availability statement:** The atomic coordinates of the human P2X3 receptor structures have been deposited in the Protein Data Bank under accession codes 21FG (ATP- and sivopixant-bound, closed) [http://doi.org/10.2210/pdb21FG/pdb] and 21DX (ATP-bound, desensitized) [http://doi.org/10.2210/pdb21DX/pdb], respectively. Cryo-EM maps were deposited in the Electron Microscopy Data Bank (EMDB) under accession codes EMD-67624 (ATP- and sivopixant-bound, closed) [https://www.ebi.ac.uk/pdbe/entry/emdb/EMD-67624] and EMD-67603 (ATP-bound, desensitized) [https://www.ebi.ac.uk/pdbe/entry/emdb/EMD-67603], respectively. All other relevant data are included in the paper or its Supporting information, including S1 Data, or have been deposited in Mendeley Data (https://doi.org/10.17632/crwytdnsdt).

**Funding:** This work was supported by funding from the National Natural Science Foundation of China (https://www.nsfc.gov.cn/english/site_1/index.html) to M.H. (32471247, 32271244, and 32411540020) and C.G. (82474171). This work was also supported by the funding of the Postdoctoral Fellowship Program of CPSF to D.W. (GZC20252599) (https://english.chinapostdoctor.org.cn/fund/fund.html) and the Open Research Fund of State Key Laboratory of Genetics and Development of Complex Phenotypes (No. SKLGDP2502, M. H.) (https://lifesupporting.fudan.edu.cn/genee/). These funders did not play any role in the study design, data collection and analysis, decision to publish, or preparation of the manuscript.

**Competing interests:** The authors have declared that no competing interests exist.

**Abbreviations:** CHS, cholesteryl hemisuccinate; CMD, conventional molecular dynamics; cryo-EM, cryogenic electron microscopy; DDM, n-dodecyl-beta-d-maltopyranoside; FBS, fetal bovine serum; MD, molecular dynamics; NAM, negative allosteric modulator; PMSF, phenylmethylsulfonyl fluoride; POPC, 1-almitoyl-2-oleoyl-sn-glycero-3-phosphocholine; SPC, simple point charge; SS, standard solution; TEV, tobacco etch virus; TM, transmembrane.

## Introduction

ATP is best known as the intracellular energy currency but also functions as an extracellular signaling molecule in many tissues via its release into the extracellular space [1,2]. P2X receptors are a family of ATP-gated cation channels that assemble as trimers from seven subtypes [3–7]. ATP binding to P2X receptors induces conformational changes that open an ion-conducting pore, while the large extracellular domain also provides multiple pockets that can be targeted by endogenous regulators and small-molecule modulators [3,4,8]. Given their widespread expression, P2X receptors contribute to diverse physiological processes, including synaptic transmission, nociception, inflammatory responses, and smooth muscle contractility [7,8].

Among P2X subtypes, P2X3 receptors are expressed in peripheral sensory neurons [9–12]. Consistent with this expression profile, P2X3 receptors have been associated with pathological roles in chronic pain and the cough reflex, motivating substantial interest in the P2X3 receptor as a therapeutic target [12,13].

Clinical proof-of-concept for targeting the P2X3 receptor in chronic cough was established with AF-219 (gefapixant), a first-generation negative allosteric modulator (NAM) that reduced cough frequency in refractory chronic cough [14]. Gefapixant has since received marketing approval for chronic cough in Europe and Japan [15,16]. However, because gefapixant also inhibits P2X2/3 heteromers, adverse taste-related events have emerged as a clinically important limitation [17,18]; accordingly, gefapixant has not been approved by the U.S. FDA [15,16]. This connection is biologically plausible because ATP is a principal transmitter from taste buds to gustatory afferents, acting through P2X2/3 receptors on sensory nerve fibers [19].

These limitations have driven the development of next-generation modulators of the P2X3 receptor with improved selectivity over P2X2/3 heteromers [20,21]. S-600918 (sivopixant), a dioxotriazine-derived clinical candidate, was optimized for high potency at the P2X3 receptor with markedly higher selectivity over P2X2/3 heteromers [22]. In a randomized clinical study in refractory chronic cough, sivopixant reduced cough frequency with a low incidence of taste disturbance, supporting the rationale that enhanced subtype selectivity may improve tolerability [23]. In addition to sivopixant, other next-generation modulators of the P2X3 receptor, such as camlipixant and eliapixant have been developed with improved selectivity over P2X2/3 heteromers and have entered clinical trials [20,24,25].

Technological advances in structural biology have provided extensive structural information on P2X receptors and have deepened our understanding of their ATP-dependent gating mechanisms [26–35], including those of P2X3 receptors. These studies have also highlighted that the orthosteric ATP-binding pocket is widely conserved across P2X subtypes, making it a poor target for subtype-selective modulation, and thereby further emphasizing the importance of NAMs for P2X receptors [4]. For other potential drug targets among the P2X receptors, particularly the P2X7 receptor, several NAM-bound structures have been reported [36–41], revealing the structural basis of their subtype selectivity. In contrast, our understanding of the action of NAMs on P2X3 receptors [41–43], especially the mechanisms underlying next-generation modulators, remains limited. Whereas intensive structural studies

and structure-based mutational analyses have been conducted for the first-generation NAM gefapixant [42,44], for sivopixant, although *in silico* simulation and corresponding mutational analyses have been reported [45], a direct experimental structure capturing sivopixant bound to the human P2X3 receptor is lacking. Although a dog P2X3 receptor-camlipixant complex structure was recently reported, structure-based mutational analysis was not performed [43], obscuring the key structural determinants of subtype specificity. Overall, despite their clinical importance, the structural basis for the subtype selectivity of next-generation P2X3 receptor NAMs remains to be elucidated.

Likewise, structural information for ternary P2X receptor-NAM-ATP complexes also remains limited, and to date only a few ternary complex structures have been reported, including the panda P2X7 (pdP2X7) receptor complex bound to A804598 and ATP and the human P2X3 receptor complex bound to compound 26a and ATP [37,46]. Among them, compound 26a binds to the TM domain and stabilizes the desensitized state [46]; therefore, its mode of action differs from that of most common P2X receptor NAMs, including A804598, which act at the allosteric site in the extracellular domain to stabilize the closed state [28]. Furthermore, in the structural analysis of the panda P2X7 receptor-A804598-ATP complex, the entire cytoplasmic domain, which is required for full channel activation, was truncated [37], and the ternary structure was obtained by the crystal soaking method [37]. These experimental conditions could influence the conformations captured in the structure; therefore, how typical NAMs of P2X receptors prevent P2X receptor activation despite ATP binding is not yet fully understood.

To address these gaps, we report the cryo-EM structure of the human P2X3 receptor in complex with ATP and sivopixant. By combining the cryo-EM structure with electrophysiology and molecular dynamics (MD) simulations, we provide structural insights into the subtype specificity and noncompetitive inhibitory mechanism of the P2X3 receptor.

## Results

### Structure determination and overall structures

We purified the human P2X3 (hP2X3) receptor protein, incubated it with sivopixant, and employed single-particle cryogenic electron microscopy (cryo-EM) to determine the structure of the hP2X3 receptor in complex with sivopixant. Cryo-EM data processing, particularly 3D variability analysis, yielded two distinct classes of EM maps (Figs 1 and S1–S3). The overall structures from the two cryo-EM maps are largely consistent and display a chalice-like trimeric assembly, in which each subunit comprises a large extracellular domain and two transmembrane (TM) helices (Fig 1). This architecture matches the canonical P2X receptor fold and shows the characteristic dolphin-like shape (Fig 1C), observed in previously reported P2X receptor structures [28].

However, as an important difference, in one map at 3.34 Å resolution, we observed residual cryo-EM density consistent with that of sivopixant and ATP (Fig 1C), whereas in the other map at 2.95 Å resolution, we observed residual density consistent with that of ATP alone (Fig 1F). Owing to the high affinity of the P2X3 receptor for ATP, ATP observed in these structures likely originated from endogenous ATP in cells that remained bound during purification [30,43].

Consistent with the cobinding of ATP and sivopixant to the P2X3 receptor in our cryo-EM structure, whole-cell patch-clamp recordings of the hP2X3 receptor showed that the sivopixant-dependent inhibition of ATP-dependent channel currents was independent of ATP concentration (Fig 3A and 3B), suggesting that its interaction with the hP2X3 receptor is independent of ATP concentration. Moreover, in our sivopixant- and ATP-bound structure, the TM domain adopted a previously-reported apo, closed conformation, with Ile323 and Thr330 forming a pore constriction [30] (Fig 1E). In contrast, in our ATP-bound structure, the TM domain adopted a conformation resembling the ATP-bound desensitized state, with Val334 forming the pore constriction [30] (Fig 1H). These observations suggest that sivopixant inhibits the channel not by stabilizing the desensitized state, but rather by stabilizing a closed state, even in the presence of ATP.

### Sivopixant binding site

In the sivopixant- and ATP-bound structure, sivopixant binds at each interface of the homotrimer within the extracellular domain (Fig 1A and 1D). In the dolphin representation, sivopixant is located in the upper body region and is

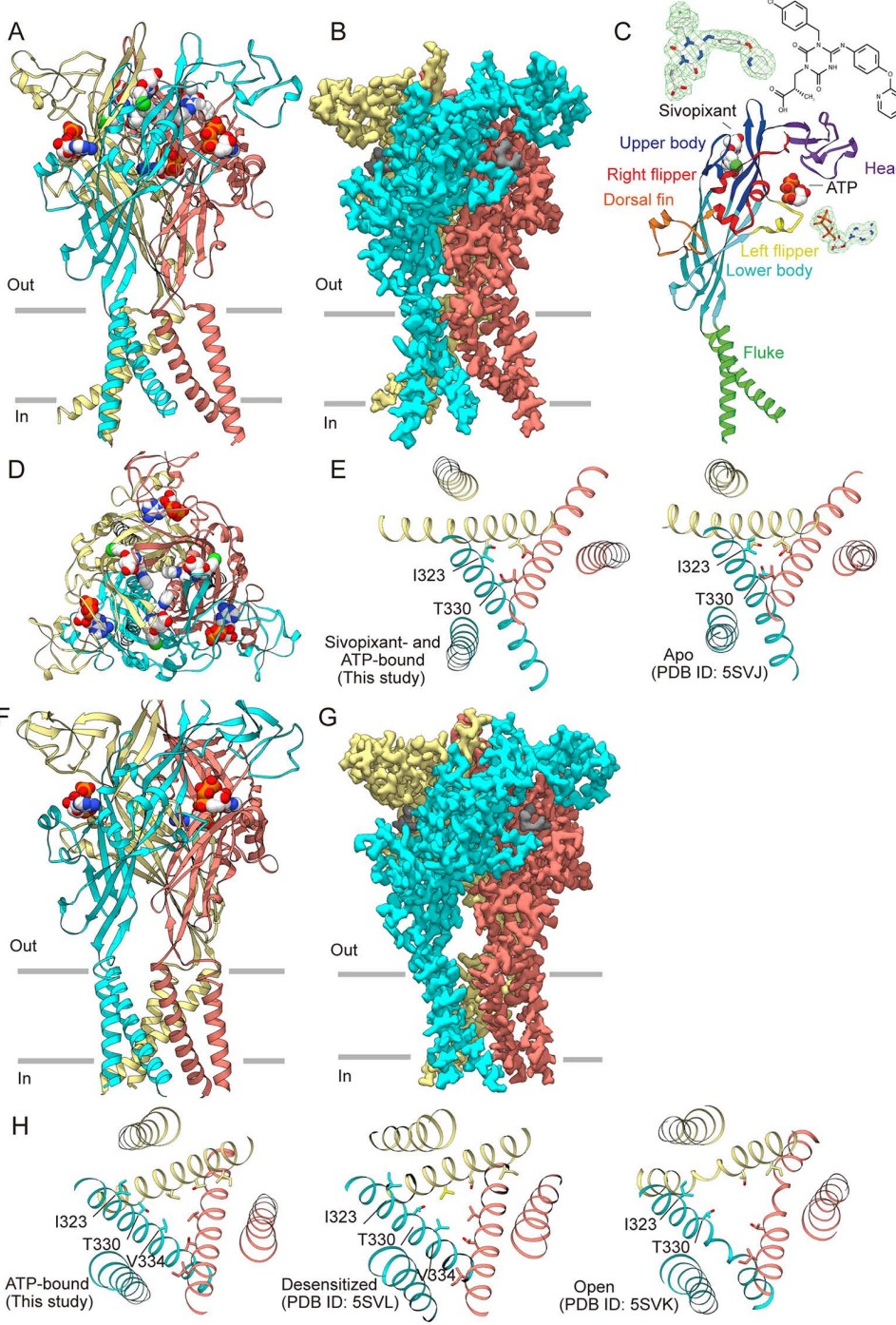

**Fig 1. Overall structures. (A, B)** Overall structure (A) and cryo-EM density map (B) of the sivopixant- and ATP-bound human P2X3 receptor viewed parallel to the membrane. Each subunit is colored distinctly. In A, the sivopixant and ATP molecules are shown in sphere representation. In B, the ligand densities are shown in gray. **(C)** The dolphin-shaped P2X3 receptor subunit is colored differently according to each structural feature. The cryo-EM densities for sivopixant and ATP are shown. **(D)** Structure of the sivopixant- and ATP-bound human P2X3 receptor viewed perpendicular to the membrane from the extracellular side. **(E)** The transmembrane domain structures of the sivopixant- and ATP-bound P2X3 receptor structure (left panel, this study) and the apo P2X3 receptor structure (right panel, PDB ID: 5SVJ) viewed from the extracellular side. **(F, G)** The overall structure (F) and cryo-EM density map (G) of the ATP-bound human P2X3 receptor viewed parallel to the membrane. Each subunit is colored distinctly. In F, ATP molecules are shown in

sphere representation, and the cryo-EM density for ATP is also shown. In G, ATP densities are shown in gray. **(H)** The transmembrane domain structures of the ATP-bound P2X3 receptor structure (this study, left panel), the desensitized P2X3 receptor structure (middle panel, PDB ID: 5SVL), and the open P2X3 receptor structure (right panel, PDB ID: 5SVK) viewed from the extracellular side.

therefore distant from the ATP-binding site (Fig 1C). Notably, a corresponding cavity space is present not only in the P2X3 receptor but also in other P2X subtypes including P2X4 and P2X7 receptors [36–40], where the cavity space serves as a hotspot for small-molecule modulation and has been referred to as the portal of the central pocket (PCP) [47].

The carboxyl group of sivopixant faces the outer side of the PCP, whereas the pyridine moiety faces the inner side (Fig 2A). Sivopixant binding is mediated mainly by hydrophobic contacts involving Met75, Ile93, Met96, Met165, Phe282, and Leu298 from one subunit and Tyr285 from the neighboring subunit (Fig 2A and 2B). In addition, the main-chain carbonyl of Asp79 and the side chain of Tyr285 form hydrogen bonds with a nitrogen atom of the triazine ring, and the side chain of Lys284 forms a hydrogen bond with the carbonyl oxygen of the triazine ring (Fig 2A and 2B). Furthermore, intersubunit salt bridges between the side-chains of Asp79 of one subunit and Arg295 of the neighboring subunit, as well as hydrogen bonds between side chains of Asp79 in one subunit and Tyr285 of the neighboring subunit, appear to stabilize the pocket architecture (Fig 2A).

Among the residues implicated in sivopixant binding, Asp79, Thr82, Met96, Lys284, and Arg295 have been suggested to be important for sensitivity to the sivopixant analog 3-(4-([3-chloro-4-isopropoxyphenyl]amino)-3-(4-methylbenzyl)-2,6-dioxo-3,6-dihydro-1,3,5-triazin-1(2H)-yl)propanoic acid (DDTPA) and/or to sivopixant itself [45]. However, for many of the additional residues that contact sivopixant in our structure, mutational validation has not yet been performed. This is likely because our previous *in silico* modeling placed DDTPA toward the outer periphery of this allosteric pocket [45] (S4A Fig), and the subsequent mutagenesis was designed on the basis of the predicted binding pose.

## Structure-based mutational analysis

We performed structure-based mutational analyses based on our experimental structure (M75A, M75W, T82I, P84A, Q85A, M96A, M96W, M165A, M165W, F282A, F282W, Y285W, and L298F) (Fig 3). We introduced alanine substitutions to attenuate side chain-mediated interactions with sivopixant. We also introduced bulkier side chain substitutions (to Trp or Phe) to create steric interference with sivopixant and thereby weaken its binding. In addition, T82I was designed to introduce a residue type found in some P2X subtypes (Figs 2A, 2C, and 4C).

First, we tested all the mutants for the inhibition of ATP-dependent channel activation by 1 μM sivopixant (Fig 3C and 3D). Compared with the wild type, M96W, M165W, and Y285W showed reduced sensitivity to sivopixant. Next, for a subset of mutants that did not show an obvious change in sensitivity to 1 μM sivopixant (M75W, T82I, P84A, Q85A, and L298F), we performed patch-clamp recordings at a lower concentration of 0.3 μM sivopixant. Under this condition, T82I exhibited a significant reduction in sensitivity (Fig 3E and 3F). Together, these structure-based mutational analyses revealed residues that contribute to sivopixant sensitivity.

## Subtype specificity

To gain structural insights into the high subtype selectivity of sivopixant, we superposed our sivopixant-bound hP2X3 receptor structure with previously-reported structures of other P2X subtypes (Fig 4). Integrating the structural comparisons with our mutational analysis results and a sequence alignment across P2X subtypes, we focused on five residues: Asp79, Thr82, Met96, Met165, and Tyr285 for further analysis (Figs 2C, 3, and 4). Among these, Asp79 has been implicated in prior mutational studies [45], whereas Thr82, Met96, Met165, and Tyr285 were identified as the residues involved in sivopixant sensitivity in our present analyses (Fig 3).

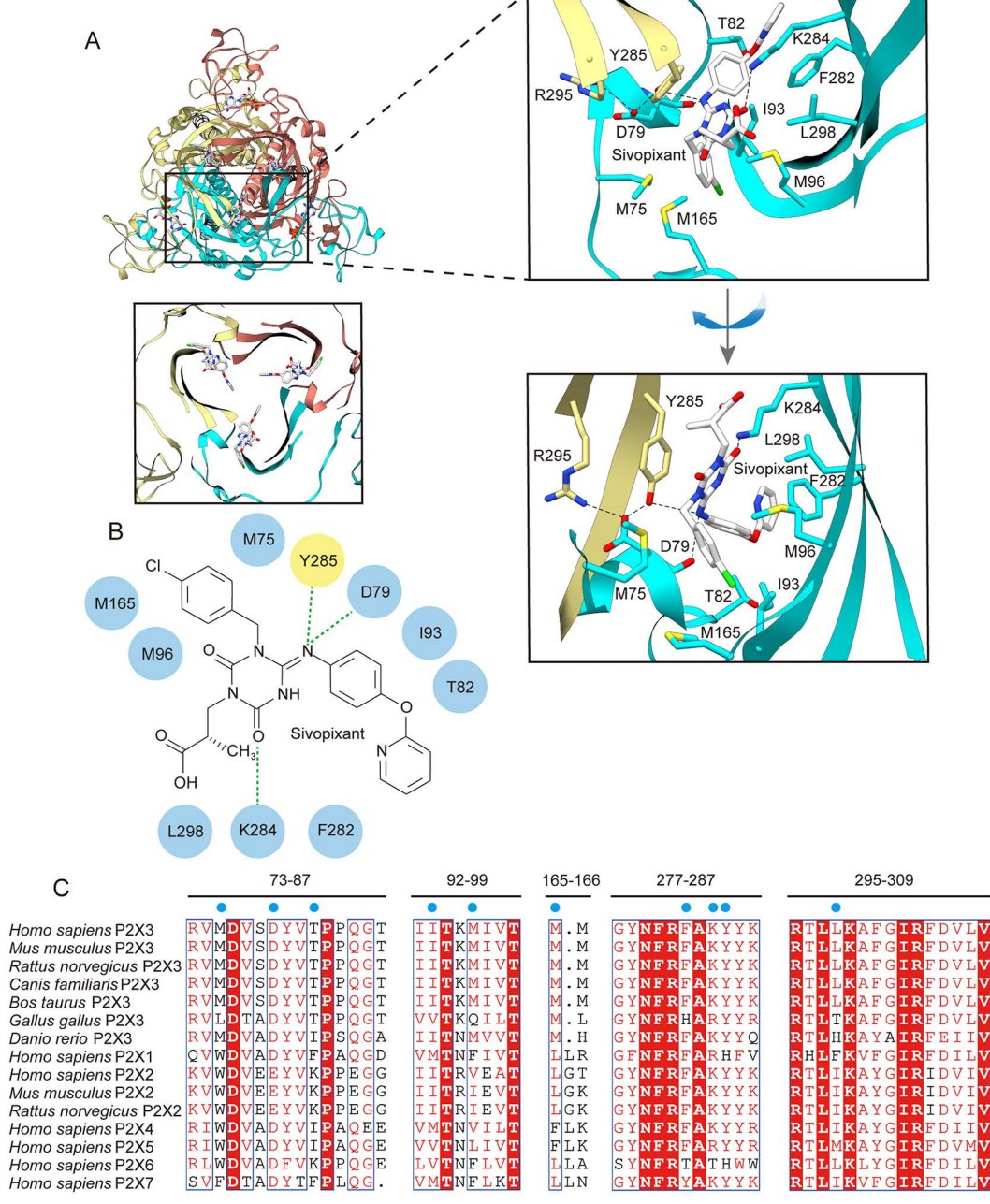

**Fig 2. Sivopixant binding site. (A)** Overall structure of the sivopixant- and ATP-bound human P2X3 receptor (upper-left panel) and a close-up view of the sivopixant binding site (lower-left panel) viewed perpendicular to the membrane from the extracellular side. Close-up views of the sivopixant binding site are shown from two different angles (upper-right and lower-right panels). The ligand molecules and amino acid residues involved in sivopixant binding are shown in stick representation. Dotted lines represent hydrogen bonds. **(B)** Schematic diagram of the interactions between the P2X3 receptor and sivopixant. Dotted lines represent hydrogen bonds. **(C)** Amino acid sequence alignment of P2X3 receptors *from Mus musculus* (Q3UR32.1), *Rattus norvegicus* (P49654.1), *Canis familiaris* (XP_038280235.1), *Bos taurus* (XP_059731161.1), *Gallus gallus* (NP_001384137.1), and *Danio rerio* (NP_571698.3); P2X receptors from *Homo sapiens* (P2X1: P51575.1; P2X2: Q9UBL9.1; P2X3: P56373.2; P2X4: Q99571.2; P2X5: Q93086.4; P2X6: O15547.2; and P2X7: Q99572.4) as well as the *Mus musculus* P2X2 receptor (Q8K3P1.2) and *Rattus norvegicus* P2X2 receptor (CAA71046.1). The residues involved in sivopixant binding are shown. The blue circles indicate the residues shown in B.

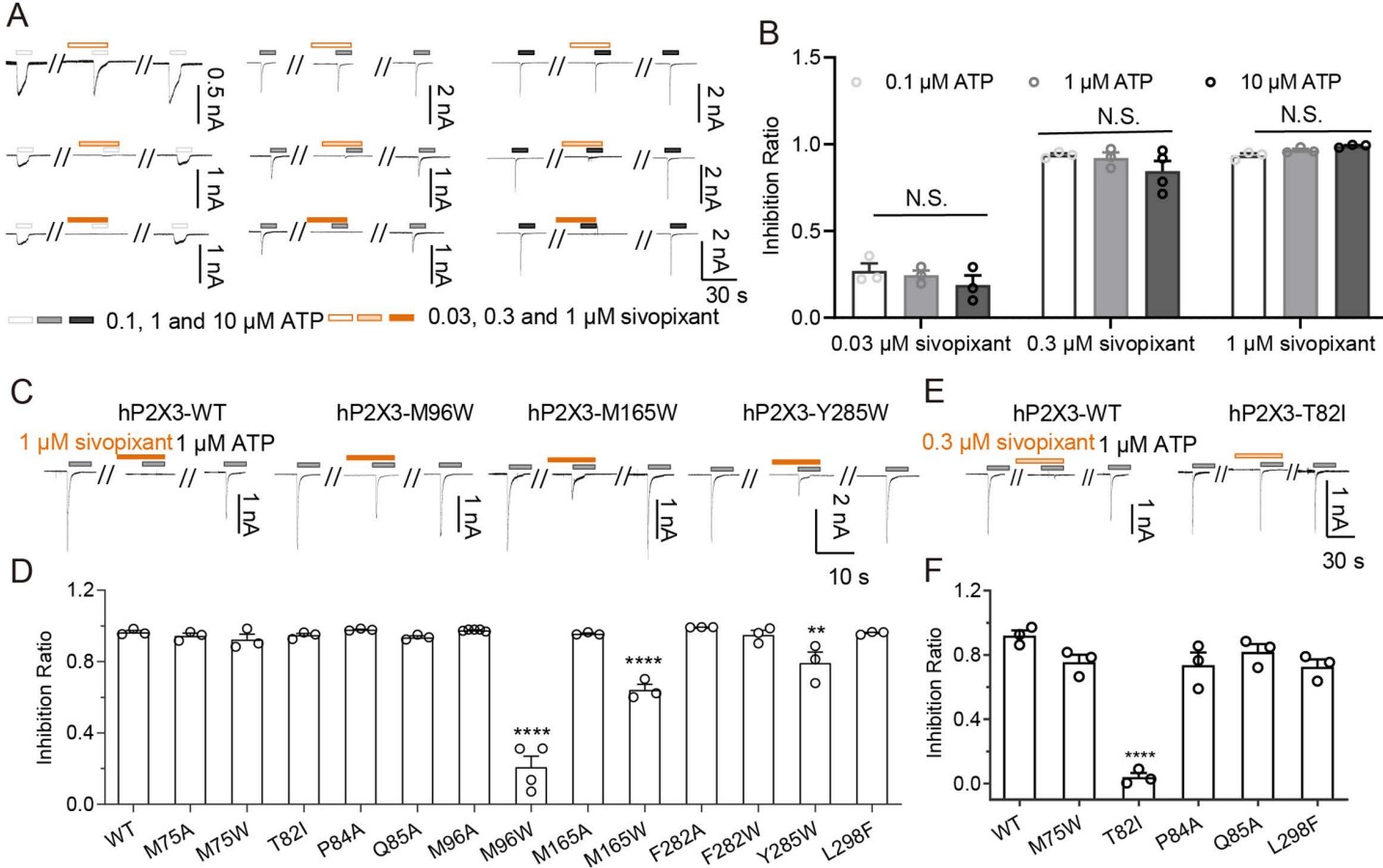

**Fig 3. Mutational analysis. (A)** Representative current traces of the effects of sivopixant on human P2X3 receptor currents at different ATP concentrations. **(B)** Effects of sivopixant on ATP (0.1, 1 and 10 μM)-evoked currents of the human P2X3 receptor (mean ± SEM, n = 3-4). **(C, E)** Representative current traces of sivopixant effects at 1 μM (C) and 0.3 μM (E) on ATP-evoked currents of the human P2X3 receptor and some mutants (C: M96W, M165W, and Y285W; E: T82I). **(D, F)** Effects of 1 μM (D) and 0.3 μM (F) sivopixant on the ATP-evoked currents of the human P2X3 receptor and its mutants (mean ± SEM, n = 3-5). Two-way ANOVA followed by Tukey's multiple comparisons test (B) and one-side one-way ANOVA followed by post hoc test **(D, F)**, \*\*$p < 0.01$, \*\*\*\*$p < 0.0001$ vs. WT). The hP2X3 WT data shown in Figs 3 and 5 were obtained from the same cells and are shown in the respective panels for comparison. The data underlying this figure can be found in S1 Data.

First, comparison of the sivopixant-bound hP2X3 receptor structure with the previously reported human P2X1 receptor structure revealed that Thr82, Met96, Met165, and Tyr285 in the hP2X3 receptor are replaced by Phe92, Phe106, Leu179, and His296 in the human P2X1 receptor, respectively (Fig 4A). Similarly, residues corresponding to Asp79, Thr82, Met96, and Met165 in the hP2X3 receptor are substituted in the P2X2 receptor by Glu97, Lys100, Val114, and Gly189, respectively (Fig 4B). In the P2X4 receptor (Fig 4C), the residues corresponding to Thr82, Met96, and Met165 in the hP2X3 receptor are Ile91, Val105, and Leu179, respectively, and in the P2X7 receptor, those residues are Phe95, Phe108, and Leu182, respectively (Fig 4D).

No experimental structures have been reported for P2X2/P2X3 heterotrimers; therefore, we generated predicted structural models of human P2X2/P2X3 heteromers with different stoichiometries (P2X2/P2X3 (2:1) and P2X2/P2X3 (1:2)) using AlphaFold3 [48]. Both models exhibited similar subunit interfaces (Figs 4E, 4F, and S5 Fig); therefore, we used the P2X2/P2X3 (2:1) model for subsequent discussion. Among the two types of subunit interfaces, one interface (P2X2-P2X3) did not differ in the corresponding region (Fig 4E), whereas the other interface (P2X3-P2X2) displayed

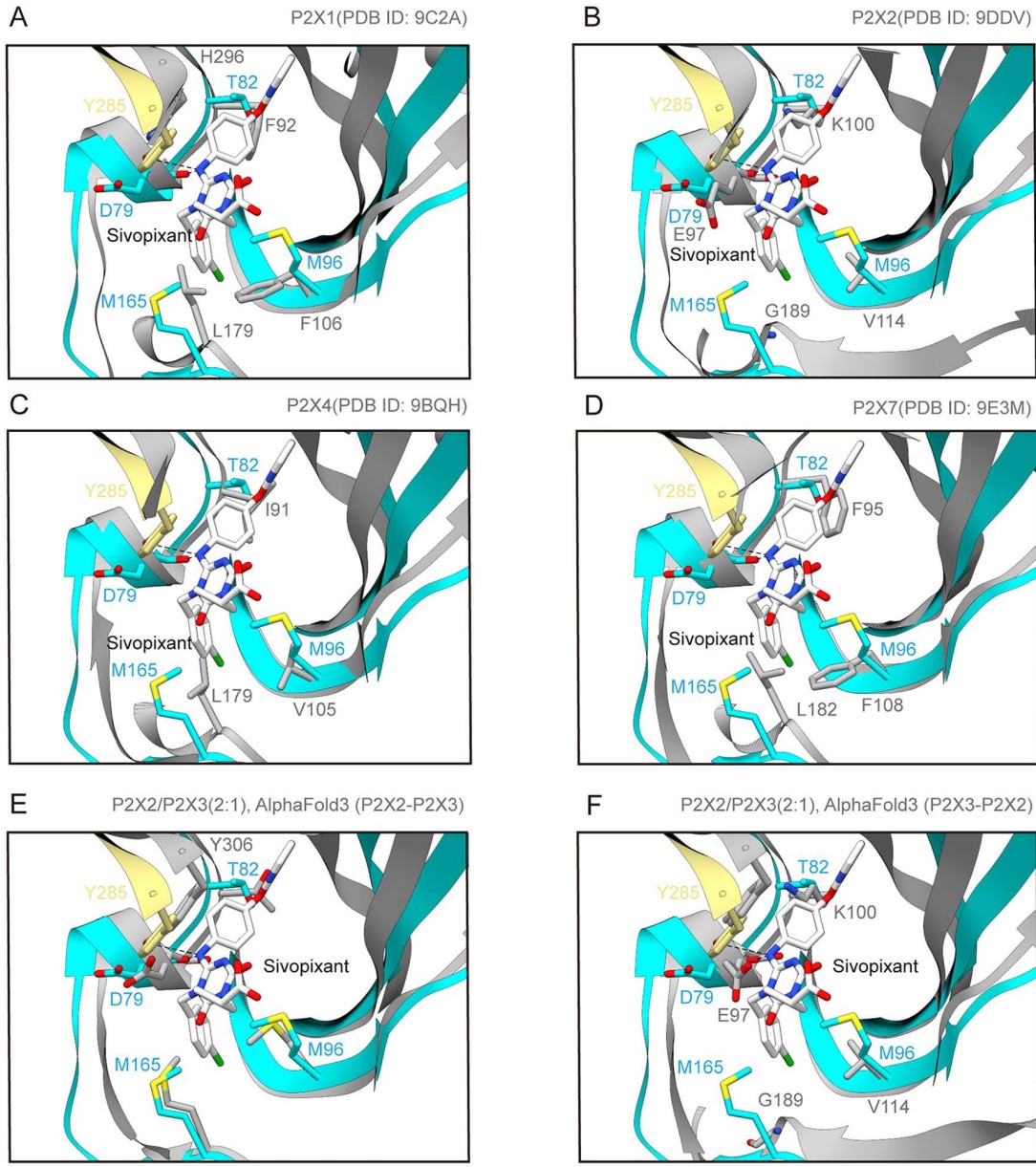

**Fig 4. Subtype specificity. (A–F)** Close-up views of the sivopixant binding site of the sivopixant- and ATP-bound P2X3 receptor structure (in this study, yellow and blue). The human P2X1 receptor structure (PDB ID: 9C2A) (A), the human P2X2 receptor structure (PDB ID: 9DDV) (B), the human P2X4 receptor structure (PDB ID: 9BQH) (C), the human P2X7 receptor structure (PDB ID: 9E3M) (D), and the predicted heterotrimer structure formed by two P2X2 subunits and one P2X3 subunit (AlphaFold3, ipTM = 0.71) (E, F) are superposed onto the P2X3 receptor structure and shown in gray. In E, the gray chain superposed onto the yellow chain is the P2X2 subunit, while the gray chain superposed onto the blue chain is the P2X3 subunit. In F, the gray chain superposed onto the yellow chain is the P2X3 subunit, while the gray chain superposed onto the blue chain is the P2X2 subunit.

clear differences derived from the P2X2 receptor, with the residues corresponding to Asp79, Thr82, Met96, and Met165 in the hP2X3 receptor replaced by Glu97, Lys100, Val114, and Gly189 (Fig 4F). Collectively, these analyses indicate that these five residues (Asp79, Thr82, Met96, Met165, and Tyr285) constitute a hotspot of intersubtype sequence divergence.

## Gain-of-function mutants

Finally, on the basis of the abovementioned insights, we designed gain-of-function (GOF) mutants of the human P2X1 and P2X2 receptors to confer sivopixant sensitivity for understanding subtype specificity (Fig 5A). Our previously designed P2X2 GOF mutant (E97D/K100T/E103Q/G105T) did not confer sivopixant sensitivity to the P2X2 homotrimer but could confer partial sensitivity to P2X2/P2X3 heteromers [45]. In this previous mutant, the targeted positions corresponding to E103Q (Gln85 in the hP2X3 receptor) and G105T (Thr87 in the hP2X3 receptor) do not directly contact sivopixant in our structure but are located near the binding pocket and could influence the shape of the pocket. In contrast, E97D (Asp79 in the hP2X3 receptor) and K100T (Thr82 in the hP2X3 receptor) correspond to two of the five residues we mentioned above (Asp79, Thr82, Met96, Met165, and Tyr285) and are positioned to directly affect sivopixant binding. The remaining three

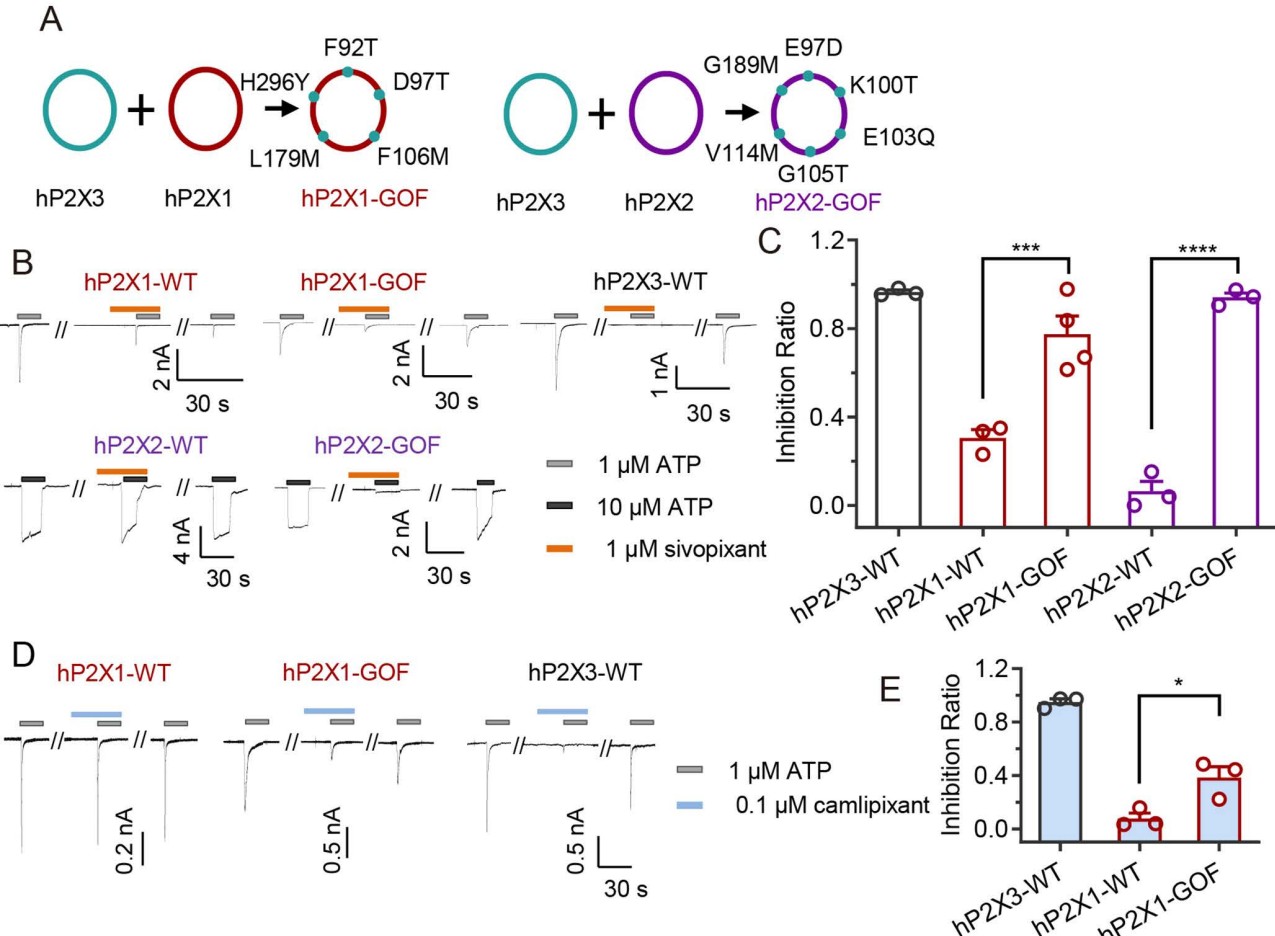

**Fig 5. Gain-of-function mutants. (A)** Schematic representation of the amino acid substitutions introduced to generate the hP2X1-GOF and hP2X2-GOF mutants. Residues in hP2X1 and hP2X2 corresponding to those in hP2X3 were replaced as indicated. **(B, D)** Representative current traces of the effects of sivopixant (B) and camlipixant (D) of on ATP-evoked currents of human P2X1, P2X2, and P2X3 wild type receptors (WT) and the gain-of-function (GOF) mutants. **(C, E)** Effects of sivopixant (C) and camlipixant (E) on ATP-evoked currents of human P2X1 and P2X3 WT receptors and the GOF mutant (mean ± SEM, $n = 3$-4). (One-way ANOVA followed by Tukey's multiple comparisons test, *$p < 0.05$ hP2X1-GOF vs. hP2X1-WT, ***$p < 0.001$ hP2X1-GOF vs. hP2X1-WT, ****$p < 0.0001$ hP2X2-GOF vs. hP2X2-WT). The hP2X3-WT data shown in Figs 3 and 5 were obtained from the same cells and are shown in the respective panels for comparison. The data underlying this figure can be found in S1 Data.

residues (Met96, Met165, and Tyr285) were not incorporated into earlier gain-of-function designs, because an experimental structure of the binding mode was not available.

Accordingly, starting from the P2X2 E97D/K100T/E103Q/G105T background, we introduced additional substitutions in the human P2X1 and P2X2 receptors to match the five key residues to their hP2X3 counterparts (hP2X1 GOF: F92T/D97T/F106M/L179M/H296Y; hP2X2 GOF: E97D/K100T/E103Q/G105T/V114M/G189M). Strikingly, whereas the wild-type hP2X2 homomer was insensitive to sivopixant, the new GOF mutant clearly displayed sivopixant sensitivity (Fig 5B and 5C). In addition, although the wild-type hP2X1 receptor exhibited only weak sensitivity to 1 μM sivopixant, the sensitivity of the GOF mutant significantly increased (Fig 5B and 5C).

In addition, we tested whether the hP2X1 and hP2X2 GOF mutants possess sensitivity to camlipixant, another P2X3-selective investigational drug. While we did not observe a significant effect of camlipixant on the hP2X2 GOF mutant even at 10 μM (S6 Fig), the hP2X1 GOF mutant showed a partially increased response to 0.1 μM camlipixant compared with the wild-type hP2X1 receptor (Fig 5C and 5D). In contrast to sivopixant, our GOF mutant showed only a partial increase in sensitivity to camlipixant. One possible explanation is that, in the previously reported camlipixant-bound human P2X3 receptor structure, Tyr70, Arg73, and Met75 in the human P2X3 receptor are involved in camlipixant binding [43], whereas our structural and mutational analyses suggested that these residues are unlikely to contribute substantially to sivopixant binding or sensitivity (Figs 2, 3, and 5). Accordingly, the corresponding residues were not included in the GOF mutants that we designed based on the sivopixant-bound structure, which may partly explain why the hP2X1 GOF mutant showed only a partial increase in sensitivity to camlipixant.

Together, these results show that the five residues implicated in our comparisons (Asp79, Thr82, Met96, Met165, and Tyr285) play a critical role in the subtype selectivity of P2X3-selective investigational drugs.

## Structural comparison

To address how sivopixant prevents activation of the P2X3 receptor despite ATP binding, we compared our hP2X3 structure in complex with sivopixant and ATP with previously reported P2X3 receptor structures in the apo, closed state and ATP-bound, open state (Fig 6). First, the ATP-binding mode in our structure is essentially the same as that in the ATP-bound, open state (Fig 6B and 6C), which is consistent with the noncompetitive action of sivopixant on the P2X3 receptor (Fig 3A and 3B). Consistent with ATP binding, we observed ATP-dependent upward motion of the dorsal fin and downward motion of the left flipper to some extent (Fig 6A), even in the presence of sivopixant. The upward motion of the dorsal fin and downward motion of the left flipper are known to be important for channel activation [49], since these motions are normally coupled with the expansion of the lower body domain, leading to channel opening (Fig 6A) [27]. However, the conformational changes in the lower body domain, particularly those near the TM domain, were largely absent in the sivopixant-bound structure (Fig 6A). Accordingly, the ion-conducting pore remains closed (Fig 1E). The lack of conformational changes in the lower body domain is likely due to sivopixant-dependent movement of the upper body domain (Fig 6A and S1 Movie). The upper body domain serves as a pivot for lower-body movements [27]; thus, expansion of the upper body may prevent it from functioning as an effective pivot, instead exerting force in the opposite direction and, thereby hindering lower-body expansion (Fig 6D and 6E and S2 Movie). This notion is consistent with the previous mutational cross-linking analysis of the upper body domain in the P2X7 receptor [50].

To test whether the structural changes in the upper body domain observed in our cryo-EM structure are dependent on sivopixant binding, we performed MD simulations using the cryo-EM structure in complex with sivopixant and ATP as the starting model, as well as models in which sivopixant or ATP was deleted, and a model in which both were deleted (Fig 7). In all MD runs, the overall structures and ligand binding were largely stable (S7 Fig). Consistent with our model, the expansion of the upper body domain was largely stable both with and without ATP throughout the MD simulations of the sivopixant-bound structures (Fig 7B and 7C), whereas the removal of sivopixant led to shorter distances between the Cα

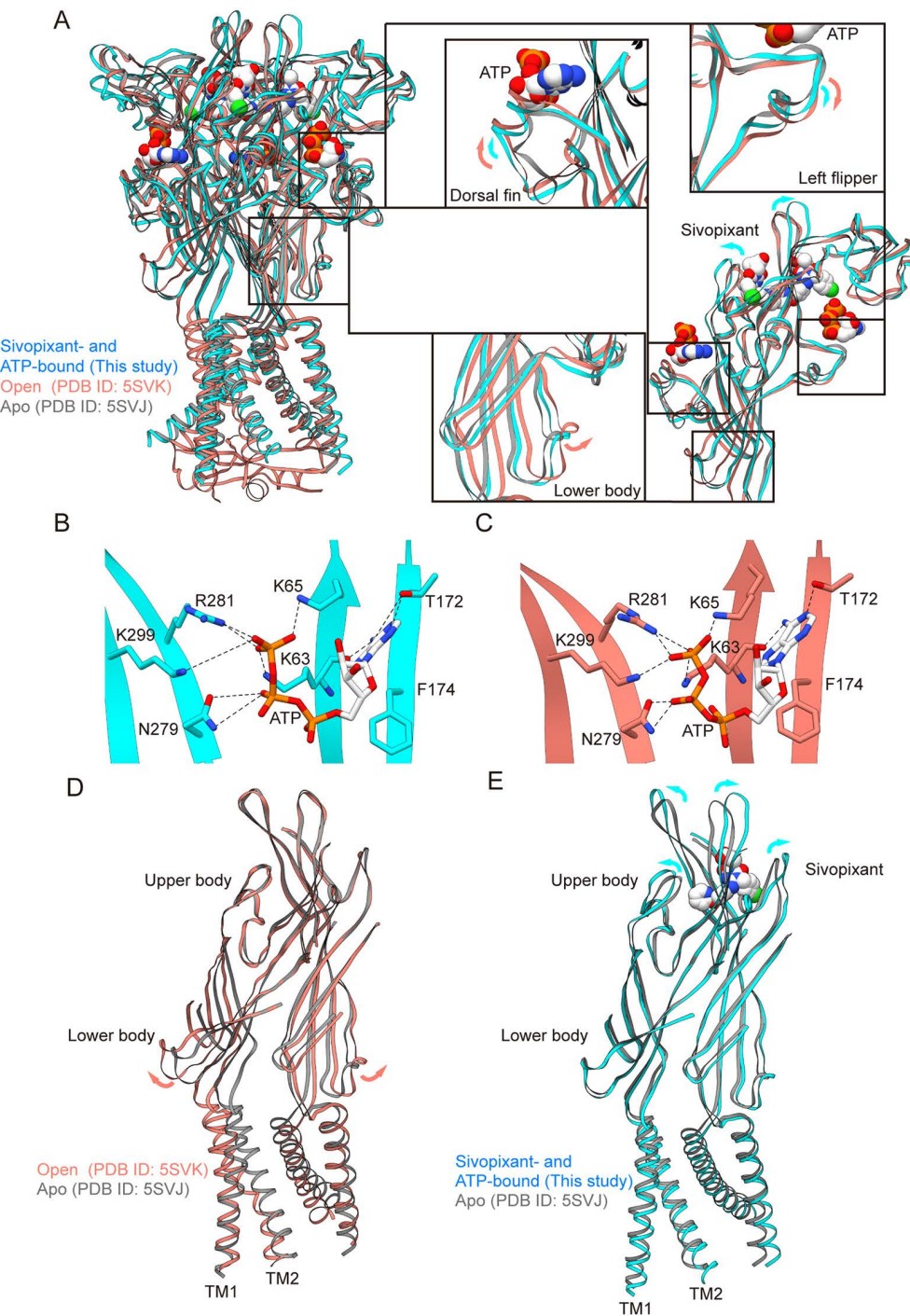

**Fig 6. Structural comparison. (A)** Superimposition of the sivopixant- and ATP-bound P2X3 receptor structure (in this study, blue) and the ATP-bound, open P2X3 receptor structure (PDB ID: 5SVK, red) onto the apo, closed P2X3 receptor structure (PDB ID: 5SVJ, gray) viewed parallel to the membrane. Close-up views of the dorsal fin, left flipper, and lower body domains are also shown. The arrows indicate conformational changes in the open P2X3 receptor structure (red) and the sivopixant- and ATP-bound P2X3 receptor structure (blue). **(B, C)** Close-up view of the ATP binding site of the sivopixant- and ATP-bound P2X3 receptor structure (in this study, blue) (B) and the open P2X3 receptor structure (PDB ID: 5SVK, red) (C). The ATP molecules and amino acid residues involved in ATP binding are shown in stick representation. Dotted lines represent hydrogen bonds. **(D, E)** Superimposition of the open P2X3 receptor structure (PDB ID: 5SVK, red) (D) and the sivopixant- and ATP-bound P2X3 receptor structure (this study, blue) (E) onto the apo P2X3 receptor structure (PDB ID: 5SVJ, gray). Only the transmembrane and body domains from the two subunits in the foreground are shown.

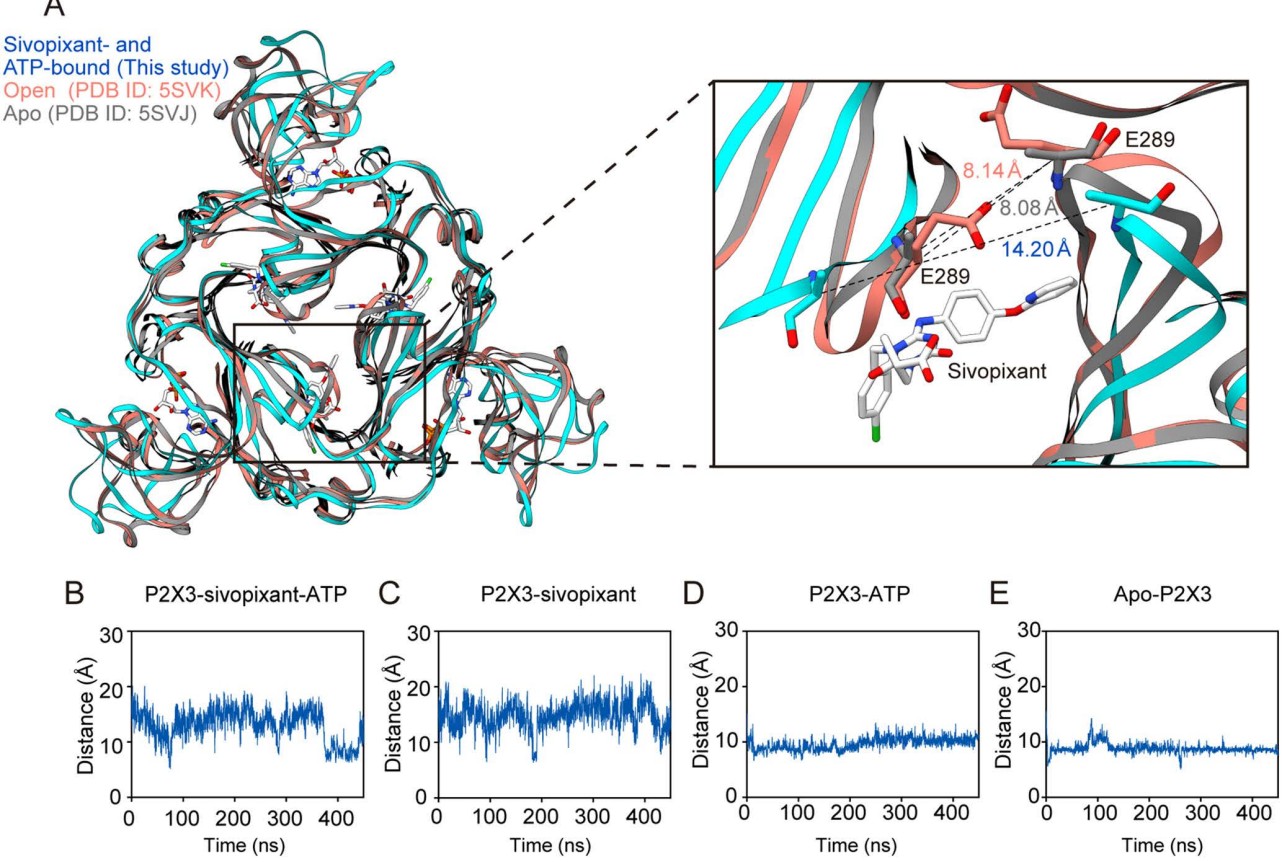

**Fig 7. MD simulations. (A)** Close-up view of the sivopixant binding site of the P2X3 receptor. Superimposition of the sivopixant- and ATP-bound P2X3 receptor structure (this study, blue) and the open P2X3 receptor structure (PDB ID: 5SVK, red) onto the apo P2X3 receptor structure (PDB ID: 5SVJ, gray) viewed perpendicular to the membrane from the extracellular side. Dotted lines indicate the distance (Å) between the Cα atoms of Glu289 in two adjacent subunits. **(B–E)** MD simulations using the sivopixant- and ATP-bound structure with both retained (B), ATP deleted (C), sivopixant deleted (D), and both deleted (E) as starting models. The distance plots of the Cα atoms between Glu289 residues of two adjacent subunits are shown. The average distances in the trimer are shown. A single MD run was performed for each condition. The data underlying this figure can be found in S1 Data.

atoms of Glu289 in the upper body domain from two adjacent subunits of the trimer, indicating a closing motion of the upper body domain in the absence of sivopixant (Fig 7D and 7E).

Taken together, these results suggest that sivopixant binding induces structural changes in the upper body domain, which in turn leads to uncoupling between the upper and lower body domains and thereby prevents channel activation even with the binding of ATP (Fig 8).

## Discussion

In this work, we determined the cryo-EM structure of the human P2X3 receptor in complex with ATP and sivopixant (Figs 1 and 2) and performed structure-based mutational analysis of the ligand binding site using patch-clamp recordings (Fig 3). The binding pose of sivopixant in our cryo-EM differs substantially from that of a sivopixant analog based on our previous *in silico* modeling [45] (S4A Fig). This discrepancy likely reflects the difficulty of predicting ligand binding starting from an apo structure, in which the allosteric site is narrower than in the ligand-bound state (S4A Fig) and therefore cannot accommodate the ligand deeply within the pocket. Consistently, when we attempted to predict the human P2X3 receptor

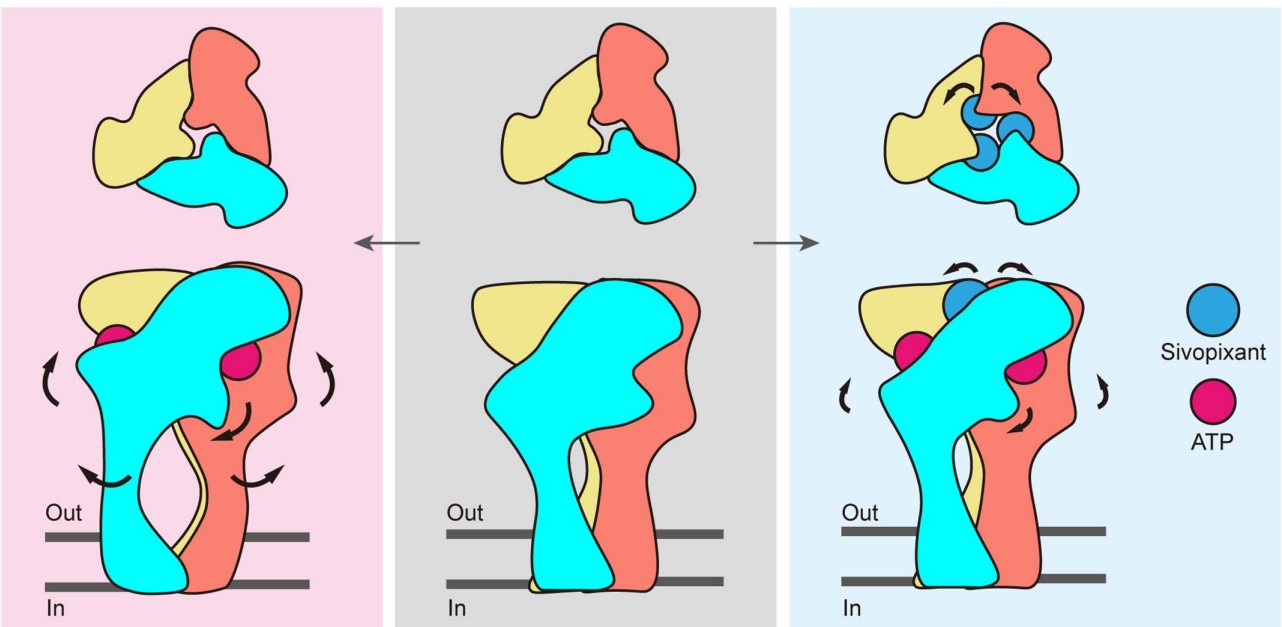

**Fig 8. Proposed noncompetitive inhibition mechanism.** Cartoon diagrams illustrating the conformational changes of the P2X3 receptor from the apo state (middle) to the ATP-bound, open state (left) and the sivopixant- and ATP-bound, closed state (right). The arrows indicate conformational changes between the states.

in complex with sivopixant using Boltz-2 [51], an AlphaFold3-derived model capable of small-molecule complex prediction, sivopixant was not placed at the allosteric site, but at unrelated sites, such as the ATP-binding pocket or the TM domain. Even when we imposed restraints on the five residues important for the subtype specificity (Asp79, Thr82, Met96, Met165, and Tyr285) in the allosteric site, the predicted pose was oriented opposite to that observed in the cryo-EM structure (S4B Fig). Together, these results show that despite recent advances in AI-based structure prediction, accurate prediction of small-molecule binding remains challenging [52].

Structural comparisons with other P2X subtypes, together with patch-clamp recordings of gain-of-function mutants, revealed key residues for subtype-specific inhibition (Figs 4 and 5). Considering that the GOF mutant conferred sensitivity not only to sivopixant but also to camlipixant, the key residues identified in this study may provide a broadly applicable structural framework for subtype selectivity of NAMs targeting the P2X3 receptor. This allosteric site corresponds to the portal of the central pocket (PCP), a hotspot for allosteric modulators across multiple P2X receptors [47]. Among previously reported P2X receptor structures in complex with subtype-selective allosteric modulators (S8A–S8D Fig), camlipixant, a selective NAM for the P2X3 receptor, has a markedly different core scaffold from sivopixant (S8A and S8B Fig). Nevertheless, superposition showed that the two compounds overlap very well in the binding site (S8E Fig). In particular, the 4-chlorophenyl group of sivopixant aligns well with the benzene portion of the benzimidazole ring of camlipixant (S8E Fig). In contrast, BAY-1797 and JNJ-54175446 are selective NAMs for the P2X4 receptor and the P2X7 receptor, respectively, and both also have substantially different chemical structures from sivopixant (S8C and S8D Fig). However, upon superposition, neither compound has a corresponding region that aligns well with the 4-chlorophenyl group of sivopixant, unlike camlipixant (S8F and S8G Fig). These observations suggest that this region may be important for conferring subtype specificity. Consistent with this idea, the 4-chlorophenyl group of sivopixant interacts with Met96 and Met165, which contribute to subtype specificity (Figs 2, 3, and 5).

Moreover, interspecies sequence differences within the PCP can cause species-dependent differences in compound sensitivity, particularly between humans and rodents, thereby complicating preclinical studies, as exemplified by the presence of Ile315 in the human P2X4 receptor and Val312 in the human P2X7 receptor [47,53]. In contrast, the residues involved in sivopixant binding are fully conserved among the human, mouse, and rat P2X3 receptors (Fig 2). Therefore, our work provides a foundation for designing compounds that are subtype selective and not affected by interspecies differences.

Our cryo-EM structure captures a ternary complex containing sivopixant and ATP; therefore, our structure also provides insight into how a NAM can inactivate the channel despite ATP binding. Structural comparisons with apo and ATP-bound open states, together with MD simulations, revealed sivopixant-dependent expansion of the upper body domain (Figs 6–8). Although we observed ATP-dependent conformational changes in the dorsal fin and left flipper (Fig 6A), which are normally associated with activation, the upper body domain, which does not directly participate in ATP binding, undergoes sivopixant-dependent expansion to suppress the expansion of the lower body domain and thereby inhibits channel opening. This inhibitory effect of lower body domain expansion on channel activation is consistent with previous mutational analysis [50]. On the other hand, in the previously reported crystal structure of the panda P2X7 receptor-NAM-ATP ternary complex [37], such NAM-dependent upper-body expansion was not observed. One possible explanation is that the experimental conditions used for structure determination, such as truncation of the entire cytoplasmic domain, which is required for full channel activation, and the soaking method, could have influenced the conformations captured in the previous structure.

Taken together, our work provides a structural basis for the subtype specificity and noncompetitive inhibition of the human P2X3 receptor, which may facilitate the rational design of next-generation P2X3 NAMs for chronic cough and other P2X3-related sensory disorders.

## Methods

### Protein expression and purification

The DNA sequence encoding the functional construct of the human P2X3 receptor for the structural studies [30] was synthesized by Genewiz (Suzhou, China) and subcloned into a pFastBac vector containing a Twin-Strep tag, an EGFP tag, and a Tobacco etch virus (TEV) protease cleavage site at the N-terminus. DH10Bac *Escherichia coli* cells were used as the host for bacmid recombination. Sf9 cells were cultured in suspension at 27 °C in SIM SF culture medium (Sino Biological, China) and routinely passaged every other day. The initial recombinant baculovirus was generated by transfecting adherent Sf9 cells with bacmid DNA using FuGENE HD reagent (Promega, USA) and was used to infect Sf9 cells for virus amplification. The EGFP-fused P2X3 receptor was overexpressed in Sf9 suspension cultures at 27 °C for 60 hours post-infection. Cells were collected and lysed by ultrasonication in TBS buffer (50 mM Tris-HCl, pH 8.0, and 150 mM NaCl) supplemented with 1 mM phenylmethylsulfonyl fluoride (PMSF), 5.2 µg/mL aprotinin, 2 µg/mL leupeptin, and 1.4 µg/mL pepstatin A. The supernatant was collected after centrifugation at 8,000*g* for 20 min and then ultracentrifuged at 185,000*g* for 1 hour. The membrane fraction pellets were solubilized in solubilization buffer [50 mM Tris-HCl, pH 8.0, 150 mM NaCl, 15% glycerol, and 2% n-dodecyl-beta-d-maltopyranoside (DDM) (Anatrace, USA)] supplemented with 1 mM PMSF, 5.2 µg/mL aprotinin, 2 µg/mL leupeptin, 1.4 µg/mL pepstatin A, and 0.2 unit/mL apyrase (Sigma, USA), and stirred for 1.5 hour. The solubilized mixture was ultracentrifuged at 185,000*g* for 1 hour. The supernatant was loaded onto Streptactin Beads 4FF beads (Smart-Lifesciences, China) pre-equilibrated with wash buffer (50 mM Tris-HCl, pH 8.0, 150 mM NaCl, 5% glycerol, and 0.05% DDM) and stirred for 1 hour. The resin was subsequently washed with 12 CV of the wash buffer. The EGFP-fused P2X3 receptor proteins were eluted with elution buffer (50 mM Tris-HCl, pH 8.0, 150 mM NaCl, 5% glycerol, 0.05% DDM, and 2.5 mM desthiobiotin), and TEV protease was added to the purified proteins to remove EGFP and the Twin-Strep tag during overnight dialysis against 1 L of dialysis buffer (150 mM NaCl, 20 mM HEPES, pH 7.5, and 0.025% DDM). The cleaved protein was applied to a Superdex 200 Increase 5/150 GL column (Cytiva, USA) pre-equilibrated with

gel filtration buffer [150 mM NaCl, 20 mM HEPES, pH 7.5, 0.001% lauryl maltose neopentyl glycol (Anatrace, USA), and 0.0001% cholesteryl hemisuccinate (CHS)]. Peak fractions were collected and concentrated to 4 mg/mL using an Amicon Ultra 50 kDa cutoff (Merck Millipore, USA). All purification processes were carried out at 4 °C.

## Cryo-EM data acquisition

For grid preparation, the P2X3 receptor was mixed with 0.1 mM Sivopixant (MedChemExpress, China). Following ultracentrifugation at 185,000$g$ for 20 min, the protein was applied to holey carbon-film grids (Quantifoil, Germany, Au R1.2/1.3 μm size/hole space, 300 mesh) that had been glow-discharged for 60 s, blotted using a Vitrobot (ThermoFisher Scientific, USA) at 100% humidity and 4 °C, and plunge-frozen in liquid ethane cooled by liquid nitrogen. Cryo-EM movies were acquired using a Falcon 4i camera (Thermo Fisher Scientific, USA) equipped on a Titan Krios (Thermo Fisher Scientific, USA) electron microscope at an acceleration voltage of 300 kV at a magnification of 130,000× for the dataset. A total of 7,464 movies were collected with a total dose of 40 electrons per Å$^2$, a pixel size of 0.959 Å, and a defocus range of −1.2 to −1.8 μm.

## Cryo-EM data processing

All data processing steps were performed in CryoSPARC [54]. Movies were motion-corrected and contrast transfer function parameters were estimated. Particles were extracted with a box size of 288 pixels, followed by two rounds of template-based particle picking and 2D classification. An ab initio reconstruction was then generated with C3 symmetry imposed. Well-defined subsets were selected and subjected to two rounds of heterogeneous refinement and nonuniform refinement [55]. After heterogeneous refinement using two 3D volumes obtained from 3D variability analysis [56], the two classes were further processed by nonuniform refinement, two rounds of local refinement, map sharpening, and local resolution estimation. The final resolution for the P2X3 receptor with ATP was 2.95 Å from 71,855 particles, whereas the final resolution for the P2X3 receptor with both sivopixant and ATP was 3.34 Å from 46,550 particles (S1 Table).

## Model building and refinement

All atomic models for the sivopixant- and ATP-bound structure and the ATP-bound structure were built in Coot [57] using previously reported P2X3 receptor structures in the apo state (PDB ID: 5SVJ) and the ATP-bound, desensitized state (PDB ID: 5SVL), respectively. After manual model adjustment in Coot, the structures were refined by real-space refinement in PHENIX [58] (S1 Table). Structure figures were generated using UCSF Chimera [59]. The sequence-alignment figure was generated using Clustal Omega [60] and ESPript 3.0 [61]. Predicted structural models of human P2X2/3 heteromers were generated using AlphaFold3 [48]. Sivopixant binding prediction was performed using Boltz-2 [51].

## Electrophysiology

The plasmid encoding the hP2X3 receptor was purchased from Open Biosystems (USA). cDNAs for hP2X1 and hP2X2 receptors were synthesized by BGI Genomics (China) and subcloned into the pEGFP-N1 vector. All mutations were generated using the KOD-Plus-Mutagenesis Kit (TOYOBO, Japan) and verified by DNA sequencing. HEK293 cells were purchased from the National Collection of Authenticated Cell Cultures (Shanghai Institutes for Biological Sciences, China), and cultured in Dulbecco's Modified Eagle Medium (Gibco, USA) supplemented with 10% fetal bovine serum (FBS) (Gibco, USA), 1% penicillin-streptomycin, and 1% GlutaMAX (Gibco, USA). Cells were cultured at 37 °C in a humidified atmosphere of 5% $CO_2$ and 95% air. Plasmids were transfected into cells using a calcium phosphate transfection reagent. Unless otherwise stated, all other compounds were purchased from Sigma-Aldrich (USA).

Recordings of currents mediated by hP2X1, hP2X2, and hP2X3 receptors were performed using a conventional whole-cell patch configuration, as previously described [62]. For conventional whole-cell recordings, the pipette solution comprised (in mM) 120 KCl, 30 NaCl, 0.5 $CaCl_2$, 1 $MgCl_2$, 10 HEPES, and 5 EGTA (pH 7.4, adjusted with

Tris-base). Specifically, hP2X3 receptor-mediated currents were recorded via perforated patch-clamp technique using nystatin to prevent current rundown. The nystatin (0.15 mg/mL) (Sangon Biotech, China) perforated intracellular solution contained (in mM) 55 KCl, 5 $MgSO_4$, 75 $K_2SO_4$, and 10 HEPES (pH 7.4, adjusted with Tris-base). HEK293 cells were recorded after 36 hours of transfection using an Axopatch 200B amplifier (Molecular Devices, USA) with a holding potential of −60 mV at room temperature (25 ± 2 °C). Current data were sampled at 10 kHz, filtered at 2 kHz, and analyzed by pCLAMP 10 (Molecular Devices, USA). HEK293 cells were bathed in standard solution (SS) containing (in mM) 2 $CaCl_2$, 1 $MgCl_2$, 150 NaCl, 5 KCl, 10 HEPES, and 10 glucose (pH 7.4, adjusted with Tris-base). ATP and other drugs were dissolved in SS and perfused immediately onto the cell membrane during the recording period via Y-tubes.

## Molecular dynamics (MD) simulations

As previously described [62,63], the energy-minimized models of hP2X3-sivopixant-ATP, hP2X3-sivopixant, hP2X3-ATP, and apo hP2X3 were used as the initial structures for molecular simulations. A large 1-palmitoyl-2-oleoyl-sn-glycero-3-phosphocholine (POPC) bilayer (at 300 K), available in the System Builder of Desmond [64], was built to generate a suitable membrane system based on the OPM database (https://opm.phar.umich.edu) [65], in which the TM domain of the hP2X3 receptor could be embedded properly. The hP2X3-sivopixant-ATP-POPC, hP2X3-sivopixant-POPC, hP2X3-ATP-POPC, and apo hP2X3-POPC systems were dissolved in simple point charge (SPC) water molecules. Counter ions were then added to compensate for the net negative charge of the system. NaCl (150 mM) was added into the simulation box that represents background salt at physiological condition. The DESMOND default relaxation protocol was applied to each system prior to the simulation run. Briefly, (1) 100 ps simulations in the NVT (constant number (N), volume (V), and temperature (T) ensemble with Brownian kinetics using a temperature of 10 K with solute heavy atoms constrained; (2) simulations were performed for 12 ps in the NVT ensemble using a Berendsen thermostat at 10 K with small time steps and solute heavy atoms constrained. (3) 12 ps simulations in the NPT (constant number (N), pressure (P), and temperature (T) ensemble using a Berendsen thermostat and barostat for 12 ps simulations at 10 K and 1 atm, with solute heavy atoms constrained; (4) 12 ps simulations using a Berendsen thermostat and Barostat at 300 K and 1 atm, with solute heavy atoms constrained; (5) 24 ps simulations using a Berendsen thermostat and Barostat at 300 K and 1 atm, with no constraints. After equilibration, the MD simulations were carried out for about 0.5 µs. The long-range electrostatic interactions were calculated using the smooth particle grid Ewald method. The trajectory recording interval was set to 200 ps and the other default parameters of DESMOND were used in the conventional molecular dynamics (CMD) simulation runs [66]. All simulations used the OPLS 2005 all-atomic force field [67–69], which is used for proteins, ions, lipids and SPC waters. The Simulation Interaction Diagram (SID) module in DESMOND [62,70] was used to explore the interaction analysis between sivopixant/ATP and the hP2X3 receptor.

All MD simulations were performed in DELL T7920 armed by NVIDIA TESLTA K40C or CAOWEI 4028GR armed by NVIDIA TESLTA K80. The simulation system was prepared, trajectory analyzed, and visualized on a CORE DELL T7500 graphics workstation with 12 CPUs.

## Supporting information

**S1 Fig. Cryo-EM analysis. (A)** Representative cryo-EM image of human P2X3 receptor particles. **(B)** Representative 2D class averages. **(C–G)** For the sivopixant- and ATP-bound human P2X3 receptor: (C) Gold-standard Fourier shell correlation (FSC) curves for resolution estimation. (D) Angular distribution of the particles used for the final map. (E–G) Side view (E), top view from the extracellular side (F), and bottom view from the cytoplasmic side (G) of the cryo-EM density map colored according to the local resolution, estimated using CryoSPARC. **(H–L)** For the ATP-bound human P2X3 receptor: (H) Gold-standard FSC curves for resolution estimation. (I) Angular distribution of the particles used for the final map.

(J–L) Side view (J), top view from the extracellular side (K), and bottom view from the cytoplasmic side (L) of the cryo-EM density map colored according to the local resolution, estimated using CryoSPARC.
(TIF)

**S2 Fig. Cryo-EM data processing workflow.** All the processing steps were performed using CryoSPARC.
(TIF)

**S3 Fig. Cryo-EM density maps for residues in the ligand-binding sites and the upper body domain. (A, B)** Close-up views of the cryo-EM density map around residues lining the sivopixant (A)- and ATP (B)-binding site. **(C, D)** Cryo-EM density maps for residues in the upper body domain in the sivopixant- and the ATP-bound structure (C) and ATP-bound (D) hP2X3 receptor structures. Residues are shown as sticks, and the cryo-EM density is shown as a green mesh.
(TIF)

**S4 Fig. Comparison between cryo-EM and predicted structures. (A, B)** Superimposition of the previously-predicted DDTPA-bound P2X3 receptor structure (A) and the Boltz-2-predicted sivopixant-bound P2X3 receptor structure (B) onto the cryo-EM structure of sivopixant- and ATP-bound P2X3 receptor viewed from the extracellular side. The cryo-EM structure is shown in surface representation (upper right panel) or in ribbon representation (lower right panel) together with the superposed predicted structures (gray). The confidence score, protein ipTM, ligand ipTM, and complex pLDDT predicted by Boltz-2 were 0.81, 0.84, 0.53, and 0.80, respectively.
(TIF)

**S5 Fig. AlphaFold3 prediction. (A, B)** The predicted heterotrimer structure formed by one P2X2 subunit and two P2X3 subunits (AlphaFold3, ipTM = 0.77) superposed onto the P2X3 receptor structure shown in gray. In A, the gray chain superposed onto the yellow chain is the P2X2 subunit, while the gray chain superposed onto the blue chain is the P2X3 subunit. In B, the gray chain superposed onto the yellow chain is the P2X3 subunit, while the gray chain superposed onto the blue chain is the P2X2 subunit.
(TIF)

**S6 Fig. Effect of camlipixant on hP2X2 wild-type and gain-of-function mutant receptors. (A)** Representative current traces showing the effects of camlipixant on ATP-evoked currents of human P2X2 wild-type (hP2X2-WT) and gain-of-function mutant (hP2X2-GOF) receptors. ATP and camlipixant were applied at 10 uM as indicated. **(B)** Inhibition ratios of camlipixant on ATP-evoked currents of hP2X2-WT and hP2X2-GOF receptors (mean ± SEM, $n = 3$–4). One-way ANOVA followed by Tukey's multiple comparisons test, N.S., not significant. The data underlying this figure can be found in S1 Data.
(TIF)

**S7 Fig. Overall structural stability during the MD simulations. (A–H)** MD simulations using the sivopixant- and ATP-bound structure with both retained (A, E, F), ATP deleted (B, G), sivopixant deleted (C, H), and both deleted (D) as starting models. The plots of the root mean square deviations (RMSDs) for Cα atoms (A–D) and the RMSD values of the atoms in sivopixant (E, G) and ATP (F, H). The data underlying this figure can be found in S1 Data.
(TIF)

**S8 Fig. Allosteric pocket of P2X receptors. (A–D)** Close-up views of the ligand binding sites of the sivopixant- and ATP-bound P2X3 receptor structure (this study) (A), the camlipixant-bound P2X3 receptor structure (PDB ID: 9BPC) (B), the BAY-1797-bound P2X4 receptor structure (PDB ID: 9BQI) (C), and the JNJ-54175446-bound P2X7 receptor structure (PDB ID: 8Z0Z) (D). The chemical structures of each compound are also shown. **(E–G)** The camlipixant-bound P2X3 (E), BAY-1797-bound P2X4 (F), and JNJ-54175446-bound P2X7 (G) receptor structures were superposed onto the sivopixant- and ATP-bound P2X3 receptor structure, and close-up views of each compound are shown in stick representation together with sivopixant.
(TIF)

**S1 Movie. Conformational changes in the sivopixant-binding pocket.**
(MP4)

**S2 Movie. Conformational changes in the body and TM domains.**
(MP4)

**S1 Table. Cryo-EM data collection, refinement, and validation statistics.**
(PDF)

**S1 Data. Numerical raw data.**
(XLSX)

## Acknowledgments

We thank the staff scientists at the Cryo-EM Facility of the School of Life Sciences, Fudan University, for technical assistance with cryo-EM data collection.

## Author contributions

**Conceptualization:** Chang-Run Guo, Motoyuki Hattori.

**Data curation:** Zhixuan Zhao, Dong-Ping Wang, Xin Zhang, Yuan Gao, Cheng Shen, Jirui Chen.

**Formal analysis:** Zhixuan Zhao, Dong-Ping Wang, Xin Zhang, Yuan Gao, Hexin Xu, Xinyu Teng, Cheng Shen, Jirui Chen, Chang-Run Guo, Motoyuki Hattori.

**Funding acquisition:** Dong-Ping Wang, Chang-Run Guo, Motoyuki Hattori.

**Investigation:** Chang-Run Guo, Motoyuki Hattori.

**Methodology:** Jinru Zhang.

**Project administration:** Chang-Run Guo, Motoyuki Hattori.

**Supervision:** Chang-Run Guo, Motoyuki Hattori.

**Validation:** Zhixuan Zhao, Dong-Ping Wang, Chang-Run Guo, Motoyuki Hattori.

**Visualization:** Zhixuan Zhao, Dong-Ping Wang.

**Writing – original draft:** Zhixuan Zhao, Dong-Ping Wang, Chang-Run Guo, Motoyuki Hattori.

**Writing – review & editing:** Zhixuan Zhao, Dong-Ping Wang, Chang-Run Guo, Motoyuki Hattori.

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
