## [Editor Report · Decision Letter 0]

11 Feb 2026

Dear Motoyuki,

Thank you for submitting your manuscript entitled "Cryo-EM reveals the structural basis of subtype-specific, noncompetitive inhibition of the human P2X3 receptor" for consideration as a Research Article by PLOS Biology. Please accept my sincere apologies for the delay in getting back to you as we consulted with an academic editor about your submission.

Your manuscript has now been evaluated by the PLOS Biology editorial staff, as well as by an academic editor with relevant expertise, and I am writing to let you know that we would like to send your submission out for external peer review.

Once your full submission is complete, your paper will undergo a series of checks in preparation for peer review. After your manuscript has passed the checks it will be sent out for review. To provide the metadata for your submission, please Login to Editorial Manager (https://www.editorialmanager.com/pbiology) within two working days, i.e. by Feb 13 2026 11:59PM.

Kind regards,

Richard

Richard Hodge, PhD

rhodge@plos.org

PLOS

---

## [Decision Letter · Decision Letter 1]

19 Mar 2026

Dear Motoyuki,

Thank you for your patience while your manuscript "Cryo-EM reveals the structural basis of subtype-specific, noncompetitive inhibition of the human P2X3 receptor" went through peer-review at PLOS Biology. Please accept my sincere apologies for the delays that you have experienced during the peer review process. Your manuscript has now been evaluated by the PLOS Biology editors, an Academic Editor with relevant expertise, and by three independent reviewers.

As you will see in the reviewer reports at the end of this e-mail, the reviewers are very positive about your study and think it is interesting and well performed. They each provide minor suggestions that we anticipate should not take you very long to address, such as changing the colour scheme in the structural figures to make them clearer and providing a supplementary figure of the cryo-EM maps. We will then assess your revised manuscript and your response to the reviewers' comments with our Academic Editor aiming to avoid further rounds of peer-review, although we might need to consult with the reviewers, depending on the nature of the revisions.

In addition, I would be grateful if you could please address the following editorial and data-related requests that I have provided below (A-F):

(A) We routinely suggest changes to titles to ensure maximum accessibility for a broad, non-specialist readership. In this case, we would suggest a minor edit to the title, as follows. Please ensure you change both the manuscript file and the online submission system, as they need to match for final acceptance:

“Structure of the human PX23 receptor reveals the basis for subtype-selective inhibition by sivopixant”

(B) You may be aware of the PLOS Data Policy, which requires that all data be made available without restriction: http://journals.plos.org/plosbiology/s/data-availability. For more information, please also see this editorial: http://dx.doi.org/10.1371/journal.pbio.1001797

-Supplementary files (e.g., excel). Please ensure that all data files are uploaded as 'Supporting Information' and are invariably referred to (in the manuscript, figure legends, and the Description field when uploading your files) using the following format verbatim: S1 Data, S2 Data, etc. Multiple panels of a single or even several figures can be included as multiple sheets in one excel file that is saved using exactly the following convention: S1_Data.xlsx (using an underscore).

-Deposition in a publicly available repository. Please also provide the accession code or a reviewer link so that we may view your data before publication.

Figure 3B, 3D, 3F, 5B, 5D, 7B, S5A-H

(C) Thank you for providing the structural data in the PDB (21FG, 21DX) and EMDB (EMD-67624, EMD-67603) databases. However, we note that the data is currently on hold for release. We ask that you please make the structures publicly available at this stage before publication.

(D) Please also ensure that each of the relevant figure legends in your manuscript include information on *WHERE THE UNDERLYING DATA CAN BE FOUND*, and ensure your supplemental data file/s has a legend.

(E) Per journal policy, if you have generated any custom code during the course of this investigation, please make it available without restrictions. Please ensure that the code is sufficiently well documented and reusable, and that your Data Statement in the Editorial Manager submission system accurately describes where your code can be found. More information on our Code Policy, what and how to share can be found here: https://journals.plos.org/plosbiology/s/code-availability

(F) Please ensure that your Data Statement in the submission system accurately describes where your data can be found and is in final format, as it will be published as written there.

**IMPORTANT - SUBMITTING YOUR REVISION**

*Resubmission Checklist*

*Published Peer Review*

*PLOS Data Policy*

*Blot and Gel Data Policy*

Best regards,

Richard

Richard Hodge, PhD

rhodge@plos.org

REVIEWS:

Reviewer #1: This is an interesting and high-quality study on the mechanism of action of a negative allosteric regulator of ATP-activated P2X3 receptor channels. Here the authors solve a cryo-EM structure of P2X3 receptors with sivopixant bound and carefully interrogate the impact of mutations in the binding site on the function effects of the compound on activation by ATP. They identify a number of residues that are critical for the allosteric actions of sivopixant and that help to rationalize the improved subtype selectivity of the drug. The authors also make several gain of function mutants to make P2X1 and P2X2 that greatly strengthen our understanding of the subtype selectivity. Finally, the authors use their structure together with previously available structures to provide a rationale for how sivopixant inhibits opening of P2X3 by ATP. I have only minor comments.

1) I would encourage the author to remake most of the structural figures as the colors are so intense that seeing where either ATP or sivopixant binds and the surrounding interactions are very difficult for the reader to see. I would also like to see density not only for the drug, but for the surrounding residues in the protein so the quality of the maps could be better evaluated.

2) Line 114 should cite Fig. 1C, I believe

3) Line 160 should cite several other figure panels I beleive

Reviewer #2: The manuscript by Zhixuan Zhao and colleagues investigates the structural basis for negative allosteric modulation of the human P2X3 receptor by the modulator sivopixant. This includes determining cryo-EM structures of P2X3 bound with ATP and a co-bound ATP and sivopixant structure. Structural findings are validated through mutagenesis and electrophysiology studies, which include gain-of-function mutations to help understand the selectivity of sivopixant along with molecular dynamic simulations. Overall, this is an extremely well written and well-presented manuscript. Nearly, all the concerns I had were addressed throughout the manuscript, which was refreshing to see.

The overall findings of this work point towards a subtle mechanism of negative allosteric modulation the include dynamic movements of the upper body of the ECD. I believe these findings would be of general interest for those in the P2X biology field and those involved in understanding allosteric modulation of proteins by ligands.

Minor concerns.

* My only real suggestion might be the inclusion of a supplemental figure that shows the cryo-EM map around these regions of difference (in the upper body) to show that there is sufficient data to support the molecular modelling to allow a fair the comparison of structures.

* Did the authors try camlipixant at the hP2X2-GOF (while I don't think is necessary to perform, it felt like an odd omission).

* Do the authors know how the binding affinity / potency of sivopixant changes based on the presence of absence of ATP? For example, is ATP required for sivopixant binding, and/or does the result in higher or lower potency for sivopixant?

Reviewer #3: This is a valuable, well performed and well presented study. It will be useful for understanding the P2X3 selectivity of NAMs versus the P2X2/X3 heteromeric receptor.

There are a number of smalle points to be addressed.

1. Title: is too general. Add: "by Sivopixant".

2. It is disturbing that the authors use "P2X3" as an abbreviation for P2X3 receptor in many places. If you want to use an abbreviations, use P2X3R (and define it before use). This has to be corrected throughout.

3. p. 4, lines 79-80: For other potential drug targets among the P2XRs, particularly the P2X7R, several NAM-bound structures have been reported. (Change the sentence. The current sentence is an exaggeration).

4. Please show chemical structures of all discussed compounds, and compare them.

5. p. 5, line 92: There are more structures with ATP, e.g. Shi et al. Sci. Adv. 2024; Kim et al., JMC 2024; etc.

6. p. 5, line 108: Fig. 2 is wrong here.

7. p.5, line 92: Several ATP/NAM-bound structures have been published, e.g. Shi et al., 2024; Kim et al., 2024 ...

Various points

p. 6, line 130: …"sivopixants binds at each interface of the homotrimer…" (please correct)

p. 7. Line 137 typo Sivopixant

p. 7, line 142, 143: side chain Asp79 (insert side chain)

p. 7, line 150: citation missing ("our previous…")

p. 8, line 160: not "commonly", only found in some…

p. 10: Is there an explanation why the GoF mutant of P2X1R changes behavior towards sivopixant and only marginally for camlipixant? Compare with published structure, Thach et al., JBC 2025).

Again, it would be helpful to show all chemical structures of the antagonists and compare them and their interactions.

p. 11: ATP-bound vs. non-ATP-bound structures with NAM: Do you expect that ATP changes the interactions?

p. 14, line 303: The clinical candidates are already optimized! How could they be further optimized? Any suggestions?

p. 15 line 330: Please check the procedure. Is it just dialysis or cleavage? Volume to perform dialysis?

p. 23, line 488: some mutants (only the mutants that show a significant change are shown).

p. 24 Figure 5: add the mutantions introduced for GoF.

p. 25, Fig. 7: how many replicates?

References: Many references cited in the Introduction are rather old. Please update.

Formatting of many of the references requires editing/correction.

Figure 1: Chemical structures in Fig. 1C are a bit too small.

Figures: Consider optimizing the coloring schemes.

---

## [Editor Report · Decision Letter 2]

9 Apr 2026

Dear Motoyuki,

On behalf of my colleagues and the Academic Editor, Raimund Dutzler, I am pleased to say that we can accept your manuscript for publication, provided you address any remaining formatting and reporting issues. These will be detailed in an email you should receive within 2-3 business days from our colleagues in the journal operations team; no action is required from you until then. Please note that we will not be able to formally accept your manuscript and schedule it for publication until you have completed any requested changes.

PRESS

Best wishes,

Richard

Richard Hodge, PhD

rhodge@plos.org

PLOS
